# Thyroid Hormones Regulate Postprandial Glucose Metabolism by Regulating SGLT1 Expression in the Small Intestine in Rats and Mice

**DOI:** 10.3390/ijms26188854

**Published:** 2025-09-11

**Authors:** Shunichi Matsumoto, Satoshi Yoshino, Shuichi Okada, Kazuhiko Horiguchi, Koshi Hashimoto, Eijiro Yamada

**Affiliations:** 1Department of Internal Medicine, Division of Endocrinology and Metabolism, Gunma University Graduate School of Medicine, Maebashi 371-8511, Japan; smatsu@gunma-u.ac.jp (S.M.); syoshino@gunma-u.ac.jp (S.Y.); okadash1823@gmail.com (S.O.); k-hori@gunma-u.ac.jp (K.H.); 2Department of Diabetes, Endocrinology and Hematology, Dokkyo Medical University Saitama Medical Center, Saitama 343-8555, Japan; k-hashi@dokkyomed.ac.jp

**Keywords:** postprandial, hyperglycemia, diabetes, thyroid hormones

## Abstract

Hyperthyroidism is known to increase basal metabolism and glucose uptake in the skeletal muscles while promoting gluconeogenesis in the liver. However, the specific mechanism underlying thyroid hormone-induced postprandial hyperglycemia remains unclear. This study explored the influence of thyroid hormones on sodium/glucose cotransporter 1 (SGLT1) expression in the small intestine and their impact on postprandial glucose metabolism. Specifically, we examined the distribution of thyroid hormone receptors in the small intestine and the subsequent effect of thyroid hormones on SGLT1 expression using rat and genetically modified mouse models. Our results demonstrated a significant upregulation of SGLT1 in the distal small intestine following T4 treatment, which corresponded with the enhanced postprandial glucose levels after oral glucose administration but not intraperitoneal administration. Furthermore, in TRβΔ337T knock-in mice that exhibited resistance to thyroid hormones, we observed increased SGLT1 expression and postprandial hyperglycemia, reinforcing our findings in rats. These findings suggest that thyroid hormones enhance glucose absorption in the small intestine via SGLT1, contributing to postprandial hyperglycemia. This study elucidates a previously unexplored aspect of thyroid hormone physiology and highlights the regulatory role of thyroid hormones in SGLT1 expression, offering potential therapeutic avenues for managing postprandial hyperglycemia in patients with diabetes.

## 1. Introduction

Reducing the risk of cardiovascular events in patients with diabetes necessitates the correction of postprandial hyperglycemia and normalization of blood glucose levels [1,2,3]. However, challenges such as the risk of hypoglycemia make it difficult to achieve adequate correction in some cases [4]. Sodium/glucose cotransporter 2 (SGLT2) inhibitors offer a means of managing blood glucose while avoiding hypoglycemia. Their mechanism of action involves the inhibition of glucose reabsorption through SGLT2 in the proximal renal tubules [5,6]. However, there are many uncertainties regarding the impact of SGLT1, also known as Slc5a1 (solute carrier family 5, sodium/glucose cotransporter, member 1, Gene ID: 20537), on systemic blood glucose, even though isoforms (1 and 2) of SGLT exist [7]. Moreover, owing to the adverse effect of severe inhibition of intestinal SGLT1, causing diarrhea, current SGLT2 inhibitors have reduced selectivity for SGLT1 [8,9].

Thyroid hormones regulate glucose metabolism; however, their mechanisms are not fully understood. In thyrotoxicosis, peripheral basal metabolism is heightened, and skeletal muscles show increased glucose uptake through upregulated glucose transporter protein type-4 (GLUT4) expression [10,11]. Conversely, the liver promotes gluconeogenesis, and the pancreas increases glucagon levels, thus elevating blood glucose levels [12,13]. More importantly, the digestive tract exhibits increased glucose absorption, leading to pronounced postprandial hyperglycemia [14,15]. This is a distinctive reaction, as demonstrated by the comparable blood glucose fluctuations in healthy individuals and those with thyrotoxicosis during intravenous glucose administration [16,17]. However, the underlying mechanisms remain unknown.

Thyroid hormone receptors (TRs), specifically TRα (Gene ID: 21833) and TRβ (Gene ID: 21834), are encoded by distinct genes, *TRA* and *TRB*, respectively. These receptors exhibit different expression patterns and functions across tissues. TRα is primarily found in the heart, skeletal muscles, and gastrointestinal (GI) tract, whereas TRβ is mainly expressed in the liver, kidneys, and brain. This differential expression indicates their specialized roles in various physiological processes [18]. Studies using genetically modified mice, such as knock-out and knock-in models, have been crucial in understanding these roles. For instance, TRα knock-out mice show abnormalities in heart rate, body temperature regulation, and GI function, whereas TRβ knock-out mice display altered cholesterol metabolism, hearing impairments, and resistance to thyroid hormone feedback regulation [19,20,21]. In the GI tract, TRα is particularly important, as confirmed by clinical observations of patients with thyroid hormone resistance α (THRα) syndrome [22,23]. These patients often experience symptoms such as constipation, growth retardation, and developmental delays, mirroring the phenotypes observed in THRα knock-out mice. Thus, both animal model studies and clinical cases highlight the critical role of TRα in gastrointestinal physiology and development.

Based on duodenal biopsies, SGLT1 reportedly increases threefold at the mRNA level and 4.3 times at the protein level in insulin-independent patients with diabetes compared to that in healthy individuals [24,25]. Additionally, impaired glucose tolerance and early-onset type 2 diabetes have been associated with elevated FT3 and FT4 levels within normal testing criteria [26]. Although studies using human colon cancer-derived cultured cells (Caco-2 cells) have shown that T3 treatment upregulates SGLT1 gene expression at the mRNA level [27], it remains unclear whether this regulatory mechanism occurs in vivo and contributes to systemic glucose homeostasis. To address this gap, we employed three complementary in vivo models. First, T4-treated rats were selected because their larger body size facilitates repeated blood sampling, enabling reliable evaluation of postprandial glucose excursions following acute thyroxine administration. Second, mice treated with T4 for 14 days served as a model of sustained pharmacological thyrotoxicosis, enabling assessment of chronic hormone exposure. Third, TRβΔ337T knock-in mice, which exhibit receptor-mediated hyperthyroxinemia while maintaining TRα function in the gastrointestinal tract, serve as a unique genetic model for investigating receptor-specific regulation of intestinal SGLT1. By integrating these pharmacological and genetic approaches across species, we sought to enhance the validity and generalizability of our findings and to clarify the physiological relationship between thyroid hormones and SGLT1 expression. This study was therefore designed to elucidate the relationship between thyroid hormones and SGLT1 using these in vivo models.

## 2. Results

### 2.1. Thyroid Receptors in the Small Intestine: Distribution and Impact on SGLT1 Expression

Previous studies have demonstrated the distribution of TRs in the liver, heart, and pituitary glands [28,29]. To explore whether thyroid receptors are also present in the small intestine and to determine the predominantly expressed isoforms, we initially investigated the tissue distributions and isoform prevalence of thyroid receptors. Our findings revealed the expression of TRs in the small intestine, with the alpha subunit of thyroid receptors being predominantly expressed in this tissue (Figure 1A). *SGLT1* is predominantly expressed in the small intestine [30,31]. Building upon our discovery of TR expression in the small intestine, we proceeded to examine the impact of thyroid hormone on *SGLT1*. As shown in Figure 1B, T4 treatment induced a hyperthyroid state in rats, resulting in an upregulation of *SGLT1* expression in the anal sites of the small intestine, with no discernible effect in the oral site (Figure 1C).

### 2.2. Thyroid Hormones Influence SGLT1 Expression and Postprandial Glucose Metabolism in the Small Intestine

We demonstrated the expression of thyroid receptors and *SGLT1* in the small intestine and showed that thyroid hormones could influence the expression of *SGLT1*. SGLT1 regulates postprandial hyperglycemia by facilitating glucose absorption in the small intestine. Therefore, we examined the role of thyroid hormones in glucose metabolism. We confirmed that the serum-free T4 levels increased following T4 administration (Figure 2A). As depicted in Figure 2B,C, intraperitoneal injection of glucose did not impact glucose metabolism under hyperthyroid conditions. Conversely, oral ingestion of glucose significantly elevated the glucose levels (*p* < 0.05) (Figure 2D,E), accompanied by significantly higher levels of *SGLT1* expression (*p* < 0.05) (Figure 2F).

### 2.3. Role of Thyroid Hormones in Postprandial Hyperglycemia: Insights from the TRβΔ337T Knock-In Mouse Model

To confirm whether thyroid hormones can regulate postprandial hyperglycemia by facilitating glucose absorption in the small intestine through *SGLT1* expression, we employed another model of hyperthyroidism. The TRβΔ337T knock-in mice have been reported to exhibit resistance to thyroid hormones [32], resulting in elevated levels of T4 (Figure 3A). As depicted in Figure 3B, there was no significant difference in the glucose levels of intraperitoneally injected mice compared to those in wild-type (WT) mice. In contrast, the area under the curve (AUC) was not different (Figure 3C). Intriguingly, postprandial hyperglycemia was observed in TRβΔ337T knock-in mice following oral ingestion compared to that in WT mice (Figure 3D), whereas the AUCs were not different (Figure 3E), attributed to an increase in *SGLT1* expression (Figure 3F). Moreover, *SGLT1* expression in TRβΔ337T knock-in mice was higher in the anal site than in the oral site (Figure 3G), confirming our findings in rats (Figure 1C).

## 3. Discussion

This study provides valuable insights into the intricate relationship between thyroid hormones and *SGLT1* expression in the small intestine, highlighting the role of thyroid hormones in postprandial hyperglycemia and potential implications for diabetes management. Our investigation corroborates previous findings regarding the distribution of *TRs* in various tissues, extending this knowledge to include the small intestine. Notably, we identified a predominant expression of the TRα subunit in the tissue, indicating its potential regulatory role in local physiological processes.

In this study, we focused on the influence of thyroid hormones on *SGLT1* expression. We observed a significant upregulation in *SGLT1* expression in the small intestine following T4 treatment, particularly in the lower segments. In the small intestine, both SGLT1 and GLUT2 are involved in glucose absorption; however, SGLT1 plays a more crucial role in rapid glucose uptake than GLUT2 [33]. Furthermore, although SGLT1 activity is downregulated in the upper small intestine under high-glucose diets or free-feeding conditions, its activity in the lower small intestine remains constant regardless of the feeding conditions, making it more susceptible to inhibition by phloridzin, a specific SGLT1 inhibitor [34]. In this context, our study identified that thyroid hormones particularly regulated *SGLT1* expression, predominantly in the lower small intestine, thereby supporting this finding.

Previous in vitro research has demonstrated that T3 treatment upregulates SGLT1 expression in intestinal epithelial (Caco-2) cells [27]. Consistent with this, our in vivo findings in rats and mice revealed that thyroid hormone excess markedly increased intestinal SGLT1 expression, thereby extending the earlier cell-based observations into a physiological context. Moreover, thyroid hormones are also known to activate Na-K-ATPase in cultured cells [35], influencing Na+ flux and potentially impacting SGLT1 activity [13]. Our in vivo study on the small intestine revealed that the effect of thyroid hormones on *SGLT1* expression varies by the intestinal segment. This discovery highlights the differential impact of thyroid hormones along the small intestine. Additionally, SGLT1 is reportedly expressed in L-cells, located in the lower small intestine, which secrete incretins such as GLP-1 and are essential for glucose-stimulated incretin release [36]. Thyroid hormones reportedly affect GLP-1 levels and response times [37,38,39,40]. This finding underscores the intricate interplay between thyroid hormones and the glucose metabolism, highlighting the role of SGLT1 in regulating postprandial hyperglycemia.

Our investigation extends beyond observational studies to elucidate the functional consequences of thyroid hormone-induced SGLT1 upregulation. Through glucose tolerance tests, we demonstrated that oral ingestion of glucose led to a significant elevation in blood glucose levels in hyperthyroid T4-treated mice, accompanied by increased *SGLT1* expression. This highlights the pivotal role of thyroid hormones in modulating glucose absorption in the small intestine, particularly in response to dietary glucose intake. To overcome the potential confounding influence of TRβ in hyperthyroid T4-treated mice, we used TRβΔ337T knock-in mice, which exhibit receptor-mediated resistance to thyroid hormones and persistently elevated serum T4 levels. Importantly, this model retains intact TRα function in the intestine, enabling us to accurately assess TRα-mediated effects on intestinal glucose absorption and SGLT1 regulation. Despite comparable glucose levels following intraperitoneal glucose administration, TRβΔ337T knock-in mice exhibited postprandial hyperglycemia following oral ingestion, accompanied by elevated *SGLT1* expression. In this study, rats were used to assess acute postprandial responses with reliable blood sampling, whereas mice enabled genetic and chronic T4 models. This complementary use of species enabled us to capture both acute and receptor-specific effects of thyroid hormones on SGLT1.

Although we identified novel mechanistic insights into the regulation of postprandial hyperglycemia by thyroid hormones through in vivo studies, factors from organs other than the small intestine may influence glucose absorption and *SGLT1* gene expression in both hyperthyroid T4-treated mice and TRβΔ337T knock-in mice. Additionally, TRα can compensate in the absence of TRβ [41], requiring careful interpretation of each phenotype. In our study utilizing TRβΔ337T knock-in mice, we demonstrated that glucose spikes were more evident in knock-in mice, which showed high serum-free T4. However, the overall AUC during OGTT was similar between knock-in and WT mice, suggesting that although TRα-mediated regulation in the small intestine contributes to postprandial glucose excursions, compensatory mechanisms in extra-intestinal tissues may counterbalance systemic glucose handling.

A limitation of this study is the lack of assessment of insulin levels during oral glucose tolerance tests. Therefore, we cannot exclude the possible contribution of altered insulin dynamics to the observed postprandial hyperglycemia. Another limitation is that we were unable to confirm SGLT1 protein expression despite attempts with multiple commercially available antibodies. Therefore, our conclusions are currently based on mRNA expression data. Future studies employing immunohistochemistry or targeted proteomics will be required to validate these findings at the protein level. Furthermore, as this study was conducted only in rodents, species differences may limit direct extrapolation to humans. Future studies using human data will be needed to validate the significance.

In summary, our findings elucidate a previously unexplored aspect of thyroid hormone physiology, highlighting the regulatory role of thyroid hormones in *SGLT1* expression and its implications for postprandial glucose metabolism. Understanding these mechanisms may offer novel therapeutic avenues for managing postprandial hyperglycemia and improving diabetes outcomes. Further research into the precise molecular mechanisms underlying thyroid hormone–SGLT1 interactions is warranted to fully harness their therapeutic potential in diabetes management.

## 4. Materials and Methods

### 4.1. Animals

Mice of different genotypes, WT (TRβ+/+) and homozygous (TRbmut/mut) for the TRβΔ337T mutation, were used. Mice were generated as previously described [32]. The TRβΔ337T knock-in mouse exhibits elevated circulating T4 caused by receptor-mediated resistance, which is distinct from simple hyperthyroidism. Importantly, TRα function in the gastrointestinal tract is preserved, serving as a unique model to investigate receptor-specific mechanisms underlying intestinal SGLT1 regulation. Male TRβΔ337T knock-in mice, C57/BL6 mice, and Wistar rats were employed for the study. In each experiment, 3 to 6 mice or rats were used. All aspects of animal care were approved by the Institutional Animal Care and Use Committee of Gunma University Graduate School of Medicine (Maebashi, Gunma, Japan) (protocol code 21-043, approved on 6 October 2021). Animals were maintained on a 12 h light–dark schedule, and they were fed laboratory chow as indicated and provided water ad libitum. The rats were divided into two groups, the T4 and Sham groups, which were injected with T4 (1.5 μg of T4/100 g body weight) and vehicle, and fasted overnight, respectively. After treatment, the rats were sacrificed, and their small intestines were harvested and divided into three sections: proximal (corresponding to the upper jejunum), middle (lower jejunum), and distal (ileum). For the mouse experiment, T4 (1.5 μg of T4/100 g body weight) was injected daily for 14 d [42].

### 4.2. Thyroid Hormone Measurement

The total serum thyroxine (T4) concentration was measured using an electrochemiluminescence immunoassay (SRL, Tokyo, Japan). Figure 1B shows how the measurement of serum-free T4 was performed to validate the efficacy of T4 administration and confirm the establishment of a hyperthyroid state.

### 4.3. RNA Extraction and Real-Time Quantitative Reverse Transcription PCR

For the analysis of TR isoforms (Figure 1A), total RNA was extracted from various mouse organs (liver, heart, pituitary, and small intestine) using an RNeasy Plus Mini Kit (Qiagen, Hilden, Germany) and quantified using a DS-11 spectrophotometer (DeNovix, Wilmington, DE, USA). For the analysis of SGLT1 expression (Figure 1C, Figure 2F and Figure 3F–G), total RNA was extracted from the small intestines of rats (Figure 1C) or mice (Figure 2F and Figure 3F–G). Single-stranded cDNA was synthesized using a High-Capacity cDNA Reverse Transcription Kit (Applied Biosystems, Waltham, MA, USA). TaqMan-based quantitative PCR was performed using a StepOnePlus Real-Time PCR System with EagleTaq Universal Master Mix (ROX) (Roche, Basel, Switzerland). TaqMan Gene Expression Assay Probes (Applied Biosystems) were used for SGLT1 (Rn01640634_m1 for rat, Mm00451203_m1 for mouse), SGLT2 (Rn00574917_m1 for rat, Mm00453831_m1 for mouse), Thra (Mm00579691_m1), Thrb (Mm00437044_m1), and Gapdh (Rn01775763_g1 for rat, Mm99999915_g1 for mouse). All quantitative PCR data were analyzed using the 2^−ΔΔCt method with *Gapdh* [43]. Absolute quantification with the standard-curve method was employed to investigate the TR isoforms in each mouse organ. The concentrations of plasmids containing TRα or TRβ were determined using standard curves. The expression of *TRA* in the liver was set to 1, and relative values were calculated accordingly.

### 4.4. Glucose Tolerance Tests

OGTTs were performed on conscious, unanesthetized male mice. After overnight fasting, blood was collected from the tails of mice at time 0, and then 2 g glucose/kg body weight was administered by oral gavage. Blood routinely collected at 30, 60, 90, and 120 min post-glucose bolus was assayed for glucose with a OneTouch Verio Vue glucometer (LifeScan, Malvern, PA, USA). Intraperitoneal glucose tolerance tests were performed exactly as the OGTTs, except that the mice received their glucose challenge as an intraperitoneal injection of 2.0 g glucose/kg body wt. The AUC for blood glucose levels was calculated to evaluate glucose tolerance. The trapezoidal rule was employed to compute the AUC from blood glucose concentration–time data.

### 4.5. Statistical Analyses

Statistical analyses were performed using analysis of variance (ANOVA) and Student’s *t*-test or the Wilcoxon/Kruskal–Wallis test using JMP pro 18 (SAS Institute Inc., Cary, NC, USA). All data are expressed as mean ± standard error of the mean.

## Figures and Tables

**Figure 1 ijms-26-08854-f001:**
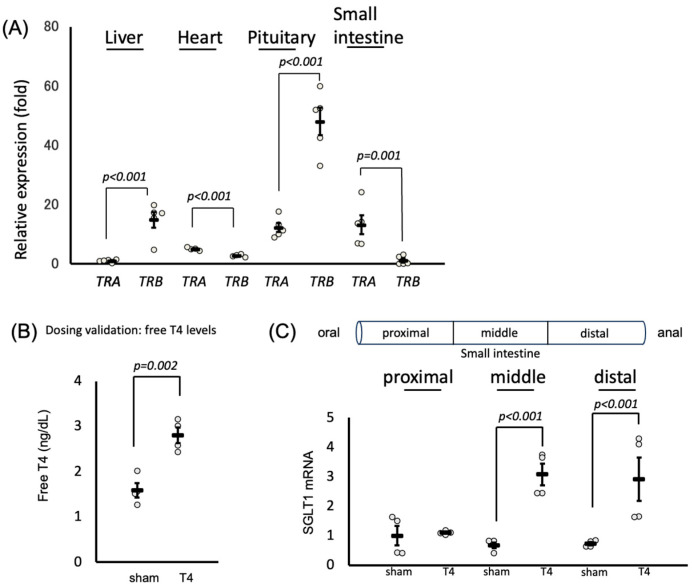
Tissue distribution of *TRs* and expression of *SGLT1* in the small intestine and other tissues. (**A**) mRNA expression levels of *TRA* and *TRB* relative to *GAPDH* in various organs (liver, heart, pituitary, and small intestine) from mice. Data are shown as mean ± SEM. (**B**) After rats were injected daily with T4 (1.5 μg of T4/100 g body weight) overnight, the total serum thyroxine (T4) concentration was measured using an ECLIA. These data are presented as dosing validation to confirm effective induction of a hyperthyroid state, rather than as mechanistic evidence. Data are shown as mean ± SEM. (**C**) Rats were injected with T4 (1.5 μg of T4/100 g body weight) overnight and then sacrificed. Their small intestines were then harvested and divided into three sections: proximal (corresponding to the upper jejunum), middle (lower jejunum), and distal (ileum), starting from the stomach side. The expression of *SGLT1* mRNAs was determined using qRT-PCR. In each experiment, four to six mice were used. Data are shown as mean ± SEM.

**Figure 2 ijms-26-08854-f002:**
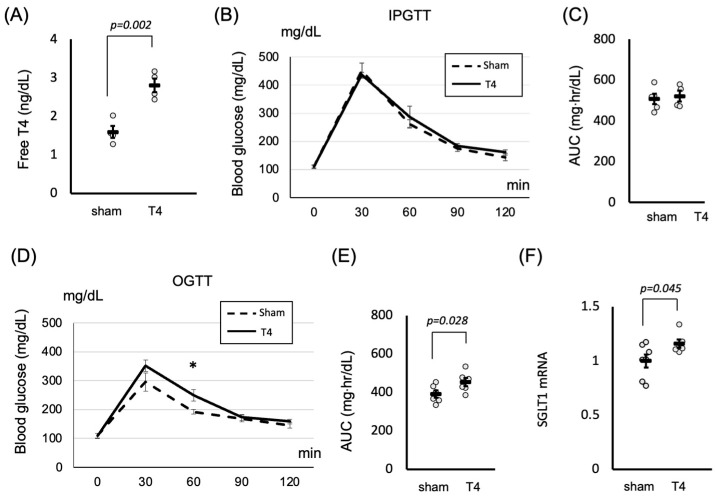
Hyperthyroid treatment uniquely elevates blood glucose levels during OGTT, attributed to heightened *SGLT1* expression. (**A**) After mice were injected daily with T4 (1.5 μg of T4/100 g body weight) for 14 days, the total serum thyroxine (T4) concentration was measured using an ECLIA. Data are shown as mean ± SEM. (**B**) After mice were injected daily with T4 (1.5 μg of T4/100 g body weight) for 14 days, IPGTT was performed, followed by glucose measurements. Data are shown as mean ± SEM. (**C**) The area under the curve (AUC) was calculated with or without T4 treatment. (**D**) After mice were injected daily with T4 (1.5 μg of T4/100 g body weight) for 14 days, an OGTT was performed, followed by glucose measurements. Data are shown as mean ± SEM, * *p* = 0.033 (**E**) AUC was calculated with or without T4 treatment. (**F**) After mice were injected daily with T4 (1.5 μg of T4/100 g body weight) for 14 days, the expression of *SGLT1* mRNAs was determined with qRT-PCR. In each experiment, four to six mice were used. Data are shown as mean ± SEM.

**Figure 3 ijms-26-08854-f003:**
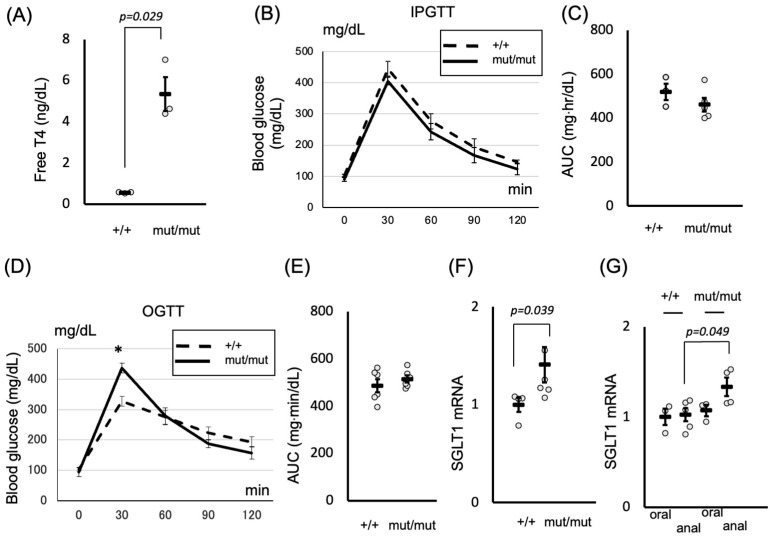
(**A**) Total serum thyroxine (T4) concentration was measured using an ECLIA. Data are shown as mean ± SEM. (**B**) IPGTT was performed, followed by glucose measurements. Data are shown as mean ± SEM. (**C**) AUC was calculated either in WT or TRβΔ337T knock-in mice. (**D**) OGTT was performed, followed by glucose measurements. Data are shown as mean ± SEM, * *p* < 0.001. (**E**) Area under the curve (AUC) was calculated in WT or TRβΔ337T knock-in mice. (**F**) The expression of *SGLT1* mRNAs was determined using qRT-PCR. Data are shown as mean ± SEM. (**G**) Small intestines were harvested and divided into two sections, A and B, starting from the stomach side. The expression of *SGLT1* mRNAs was determined using qRT-PCR. In each experiment, three to six mice were used. Data are shown as mean ± SEM.

## Data Availability

The original contributions presented in the study are included in the article; further inquiries can be directed to the corresponding author.

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
