# Peer review of "Thyroid Hormones Regulate Postprandial Glucose Metabolism by Regulating SGLT1 Expression in the Small Intestine in Rats and Mice"

_ijms, 2025, doi:10.3390/ijms26188854_

Round 1
Reviewer 1 Report (New Reviewer)
Comments and Suggestions for Authors
I have reviewed the manuscript entitled (Thyroid hormones regulate postprandial glucose metabolism by regulating SGLT1 expression in the small intestine) , submitted to (IJMS), which investigates the role of thyroid hormone in postprandial glucose metabolism, with a particular focus on SGLT1 expression in the small intestine. My evaluation considers the clarity, methodological rigor, data presentation, and interpretation of the results, as well as the alignment between the study’s aims and the evidence provided.
While the study addresses a potentially interesting link between thyroid hormone signaling and intestinal glucose absorption, several critical issues limit the current manuscript’s clarity and scientific robustness. My detailed comments, organized point-by-point, are provided below to help guide a thorough revision.
Main comment
- The abstract states that “the mechanism underlying thyroid hormone-induced oxyhyperglycemia remains unclear,” implying that this study will address the “oxy” component the link between hyperglycemia and increased oxidative metabolism. However, the experiments focus solely on thyroid hormone effects on intestinal SGLT1 expression and postprandial glucose excursions. No measurements of oxygen consumption or oxidative metabolism are included. This overextends the rationale and may mislead readers about the study’s scope.
- The study is described as “exploring the effect of elevated thyroid hormone on SGLT1 in the small intestine using genetically modified mice and rats,” yet there is no clear rationale provided for using two different species. This raises questions about comparability, particularly given known species differences in thyroid hormone metabolism and intestinal transporter regulation.
- In the Introduction, the authors cite Matosin-Matekalo et al. (Biochem J, 1998) in the context of TRα and GI tract function. However, that study specifically examined T3-induced regulation of SGLT1 in intestinal epithelial cells. The authors should draw on the relevant findings from this reference to compare their results directly to those earlier observations.
- The TRβΔ337T knock-in mouse exhibits elevated T4 due to receptor-mediated resistance, a phenotype distinct from simple hyperthyroidism. The rationale for using this model to study intestinal SGLT1 regulation should be clarified, particularly when other experiments use T4-treated rats. The combination of different species and thyroid states complicates interpretation and requires stronger justification.
- While the authors attempted SGLT1 protein detection using multiple antibodies, the inability to detect a specific band raises concerns about whether the observed mRNA upregulation translates to protein-level changes. Alternative approaches such as optimized Western blot conditions, immunohistochemistry, or targeted proteomics should be considered.
- It should also be clarified whether T4 treatment affected other key intestinal glucose transporters, such as GLUT2, or altered their localization. Additionally, measurements of gastrointestinal transit time would help exclude changes in absorption kinetics unrelated to transporter expression.
- The figures require substantial revision for accuracy and consistency, and this undermines the clarity of the Results section, making it difficult to follow the data flow and evaluate the manuscript scientifically. For example, in Figures 1 and 3, the legends describe “mean ± SEM” and refer to (open bars) and (solid bars), yet only dot plots are shown with no visible error bars. Legends should match the actual figure format, and SEM should be displayed if it has been calculated. Sample sizes (n) must be stated. In Figure 1A, the source of TRα/TRβ mRNA data is unclear and should be specified. Figure 1B’s free T4 values should be framed as dosing validation, not mechanistic evidence, and clearly labeled. Figure 1C uses vague (A/B/C) labels and anatomically incorrect terms (oral/anal side), which should be replaced with standard anatomical terms and defined in the Methods. If the data in Figure 1 are from rats, this should be clearly stated, given later use of TRβΔ337T mice. In Figure 3A, the WT group has only a single replicate (n = 1), which is inadequate for statistical comparison. These issues collectively highlight the need for consistent figure formatting, accurate legends, and a more clearly organized Results section.
Miner points
- The title should specify the species studied (e.g., “rat” or “mouse”) to avoid implying cross-species generalization.
- Details such as the electrochemiluminescence immunoassay should be described in the Methods section, not introduced for the first time in the Results.
- assessing insulin levels during the oral GTT would clarify whether postprandial hyperglycemia is driven purely by absorption or also involves altered insulin dynamics.
Author Response
Please see the attachment.

Reviewer 2 Report (New Reviewer)
Comments and Suggestions for Authors
The manuscript explores a role of a specific gene/protein in metabolism. The study is well planned and executed. Some more validation efforts would have improved the study, but they are not necessary.
Please report the exact P values, not just p<0.05. The sample sizes are reported rather unsystematically, and it is somewhat confusing to not see this being reported. One thing that would have improved the study further is a sort of replication, or at least some indication of it.
Lastly, I am not sure why you did not try to delineate this in humans, at least in part, since there are a lot of datasets with relevant data currently available, even in the free domain. I can only assume that you planned to carry this on in another study, so this is just a generalized suggestion (there may be a chance of different metabolisms in mice and humans, and this is the main worry of clinical physicians - how valid and relevant is this for humans).
The manuscript is nicely written and should be published after these few minor issues are discussed and/or improved.
Round 2
Reviewer 1 Report (New Reviewer)
Comments and Suggestions for Authors
The author has corrected the manuscript as requested. No further action is required at this time.
This manuscript is a resubmission of an earlier submission. The following is a list of the peer review reports and author responses from that submission.
Round 1
Reviewer 1 Report
Comments and Suggestions for Authors
The manuscript entitled “Thyroid hormones regulate postprandial glucose metabolism by regulating SGLT1 expression in the small intestine” by Shunichi Matsumoto et al., was presented as an Original Article belonging to the “Molecular Endocrinology and Metabolism” section (Special Issue: Metabolism and Diseases Related to Thyroid Function). From the title the authors aimed to provide evidence about the effects of Thyroid hormones on Sodium/glucose cotransporter 1 (SGLT1) by using in vivo rat and mouse models. Although the thematic of the research is relevant and the article could be of potential interest not only for a wide community of basic researchers working on endocrinological diseases but also for clinicians, in the current state the manuscript appears still preliminary, and the quality of the data does not reach the expected quality standards required for publication. Several major points need to be addressed to strengthen the technical rigor.
Comments for the text:
Globally, the manuscript needs to be improved in terms of data presentation and writing to make the work more accessible and complete to the reader. The introduction needs to be revised: the authors should provide a more detailed background and highlight the objectives, aims and novelty about this study. Concerning the gene nomenclature, where the authors referred to TRα and TRβ as genes, please replace it with own gene name, i.e. Thra (Gene ID: 21833, https://www.ncbi.nlm.nih.gov/gene/21833) and Thrb (Gene ID: 21834, https://www.ncbi.nlm.nih.gov/gene/21834). Analogously for SGLT1, whose complete and proper name is Slc5a1 solute carrier family 5 (sodium/glucose cotransporter), member 1 (Gene ID: 20537, https://www.ncbi.nlm.nih.gov/gene/20537). Moreover, gene name abbreviations must be italicized.
Comments for the data:
In general, the data presented in the figures need to be improved. One main limitation of the study is that the validation methods are not abundant enough. Therefore, to support the conclusions (“we identified novel mechanistic insights into the regulation of postprandial hyperglycemia by thyroid hormones through in vivo studies”), the results should be improved including several experimental approaches and different methodologies.
1. The qRT-PCR relative expression analysis of SGLT1 in the small intestine (Figure 1) was normalized against a single housekeeping gene, which is somewhat unusual. Have the authors tested other housekeeping genes to confirm stability of GAPDH under the treatment conditions?
2. How many animals (mice and/or rats) were used in the experimental procedures? No information is provided in Material and Methods section. For a better interpretation of the data, column graphs should be replaced by dot-plot graphs to identify the number of individual animals involved in the experimental groups.
3. Although the authors reported that treatment with T4 increases SGLT1 expression levels in the small intestine of rats, the data shown in Figure 2F do not appear to confirm a significant SGLT1 upregulation following hyperthyroid treatment. This data, not clearly evident from the qRT-PCR analyses, should necessarily be confirmed by protein expression. The same goes for the TRbD337T knock in mouse model.
4. Authors should consider carrying out a rescue experiment, making mouse and/or rat models hypothyroid by administering Methimazole and evaluating SGLT1 expression in the small intestine of mice or rats under hypothyroid conditions.
5. In vivo data should be corroborated by in vitro analyses using a small intestine cell line, treated with T4 (to mimics hyperthyroid conditions) and tT3 (to mimics hypothyroid conditions) to assess transcriptomic and protein expression of SGLT1. Furthermore, rescue experiments using a shRNA or siRNA against SGLT1 are necessary to explore the mechanistic regulatory pathway governing the thyroid hormones effects under postprandial hyperglycemia condition.
Comments for the figures:
1. Figures are not properly cited in the Results section (Fig. 2A is not cited). Make sure to correctly cite each figure in the text
Author Response
Comments for the text
Comments: Globally, the manuscript needs to be improved in terms of data presentation and writing to make the work more accessible and complete to the reader. The introduction needs to be revised: the authors should provide a more detailed background and highlight the objectives, aims and novelty about this study. Concerning the gene nomenclature, where the authors referred to TRα and TRβ as genes, please replace it with own gene name, i.e. Thra (Gene ID: 21833, https://www.ncbi.nlm.nih.gov/gene/21833) and Thrb (Gene ID: 21834, https://www.ncbi.nlm.nih.gov/gene/21834). Analogously for SGLT1, whose complete and proper name is Slc5a1 solute carrier family 5 (sodium/glucose cotransporter), member 1 (Gene ID: 20537, https://www.ncbi.nlm.nih.gov/gene/20537). Moreover, gene name abbreviations must be italicized.
Response: As per your suggestions, we have revised the text to conform to gene nomenclature guidelines. Additionally, we have italicized the gene abbreviations.
Comments for the data
Comments #1: The qRT-PCR relative expression analysis of SGLT1 in the small intestine (Figure 1) was normalized against a single housekeeping gene, which is somewhat unusual. Have the authors tested other housekeeping genes to confirm stability of GAPDH under the treatment conditions?
Response#1: This figure was quantified using absolute quantification with plasmids to measure the concentrations of both TRA and TRB, with normalization by GAPDH. The expression of TRA in the liver was set as 1, and the relative expression of TRA and TRB in various organs was analyzed. We have added this information to the Materials and Methods section for clarity.
Comments #2: How many animals (mice and/or rats) were used in the experimental procedures? No information is provided in Material and Methods section. For a better interpretation of the data, column graphs should be replaced by dot-plot graphs to identify the number of individual animals involved in the experimental groups.
Response #2: We have included the number of mice used in each experiment and have replaced the column graphs with dot plot graphs.
Comments #3: Although the authors reported that treatment with T4 increases SGLT1 expression levels in the small intestine of rats, the data shown in Figure 2F do not appear to confirm a significant SGLT1 upregulation following hyperthyroid treatment. This data, not clearly evident from the qRT-PCR analyses, should necessarily be confirmed by protein expression. The same goes for the TRbD337T knock in mouse model.
Response #3: Figure 2F shows that T4 administration significantly upregulates SGLT1 gene expression. We have enlarged the asterisk for better visibility. As mentioned in Comment #5, unfortunately, confirmation at the protein level is currently not feasible as it requires an extensive amount of time to prepare. This includes aligning the parameters for knock-in mice and verifying the appropriate antibody for SGLT1 corresponding to the Western blot. We were recommend publishing this article as a Communication. The observed increase in SGLT1 mRNA levels alone is a novel finding that could partially explain the mechanism underlying the hyperglycemia observed during OGTT in states of thyroid hormone excess. Nevertheless, we aim to explore this aspect in future studies. We request for your kind consideration in this regard.
Comments #4: Authors should consider carrying out a rescue experiment, making mouse and/or rat models hypothyroid by administering Methimazole and evaluating SGLT1 expression in the small intestine of mice or rats under hypothyroid conditions.
Response #4: While oxyhyperglycemia is observed in hyperthyroidism, hypoglycemia is typically not seen in hypothyroidism alone, suggesting a minimal effect of SGLT1. Investigating hypothyroid treatment would require significant preparation. Could you please provide your opinion on whether we should pursue this investigation?
Comments #5: In vivo data should be corroborated by in vitro analyses using a small intestine cell line, treated with T4 (to mimics hyperthyroid conditions) and tT3 (to mimics hypothyroid conditions) to assess transcriptomic and protein expression of SGLT1. Furthermore, rescue experiments using a shRNA or siRNA against SGLT1 are necessary to explore the mechanistic regulatory pathway governing the thyroid hormones effects under postprandial hyperglycemia condition.
Response #5: As you mentioned, in vitro analysis would typically be considered. However, we were recommend publishing this as a Communication and have therefore summarized the findings at the RNA level. Nevertheless, we aim to explore this aspect in future studies. We request for your kind consideration in this regard.
Comments for the figures
Comments #1: Figures are not properly cited in the Results section (Fig. 2A is not cited). Make sure to correctly cite each figure in the text.
Response #1: As suggested, we have added a citation for Fig. 2A within the text.
Reviewer 2 Report
Comments and Suggestions for Authors
In this article entitled "Thyroid hormones regulate postprandial glucose metabolism by regulating SGLT1 expression in the small intestine", the authors used T4-administered rats and TRβΔ337T knock mouse models to study the effects of thyroid hormones on SGLT1 expression and its effects on postprandial glucose metabolism.This study found some very interesting phenomena. Thyroid hormone can upregulate SGLT1. After injection of T4, SGLT1 in the distal small intestine is significantly upregulated, thereby increasing glucose absorption and leading to postprandial hyperglycemia. This upregulation is related to the increase in postprandial blood glucose levels after oral glucose administration. The authors further discussed the significance of thyroid hormone and glucose metabolism, linking the upregulation of SGLT1 to the potential mechanism of thyroid hormone-induced hyperglycemia, and emphasizing the therapeutic potential of targeting SGLT1 to treat postprandial hyperglycemia in diabetic patients.However, the study used an animal model, which is only a theoretical basis, and further exploration is needed for its application in human clinical practice. The mechanism of how thyroid hormone affects SGLT1 expression can be further explored, which are the shortcomings of this manuscript. Therefore, it needs to be major revision before it can be published in International Journal of Molecular Sciences.
1. In line 78, Our previous study demonstrated the distribution of TRs in the liver, heart, and pituitary glands [28,29]. Why is it cited from others? Is Our previous study demonstrated an incorrect statement? In addition, there are fewer references in recent years. Please update the references.
2. In line 166, Our in vivo study on the small intestine revealed that the effect of thyroid hormones on SGLT1 expression varies by the intestinal segment. This discovery highlights the differential impact of thyroid hormones along the small intestine. Since the effect of thyroid hormones on SGLT1 expression in the small intestine varies by intestinal segment, and the proximal small intestine mainly enters cells through glucose transporters (GLUTs), will thyroid hormones affect GLUTs? Moreover, the release of thyroid hormones itself promotes glucose absorption, and the authors did not point out how thyroid hormones affect SGLT1 expression after acting on receptors.
3. In line 215, T4 (1.5 μg of T4/100 g body weight), what is the basis for the concentration of T4 injection? In the body, T4 is mainly used as a reserve form of thyroid hormone, while T3 has a more direct and active effect on cells. So did the author consider the relationship between T3 and T4?
4. Hormones have negative feedback mechanisms, and hormone regulation can be regulated by multiple hormones. Did the author study or exclude the interference of other hormones on SGLT1?

Author Response
Comments #1: In line 78, Our previous study demonstrated the distribution of TRs in the liver, heart, and pituitary glands [28,29]. Why is it cited from others? Is Our previous study demonstrated an incorrect statement? In addition, there are fewer references in recent years. Please update the references.
Response #1: This was an error in our writing. The word "our" was unnecessary, as the sentence refers to previous reports. We have updated the references to include more recent publications.
Comments #2: In line 166, Our in vivo study on the small intestine revealed that the effect of thyroid hormones on SGLT1 expression varies by the intestinal segment. This discovery highlights the differential impact of thyroid hormones along the small intestine. Since the effect of thyroid hormones on SGLT1 expression in the small intestine varies by intestinal segment, and the proximal small intestine mainly enters cells through glucose transporters (GLUTs), will thyroid hormones affect GLUTs? Moreover, the release of thyroid hormones itself promotes glucose absorption, and the authors did not point out how thyroid hormones affect SGLT1 expression after acting on receptors.
Response #2: As you noted, SGLT1 on the brush border membrane and GLUT2 on the basolateral membrane are crucial for glucose absorption in the small intestine. Although the effects of thyroid hormone on GLUT2 expression in the liver and β-cells have been reported, details of its expression in the small intestine remain elusive.
While your suggestion to investigate the role of GLUT2 in the increased SGLT1 expression observed in OGTT under thyroid hormone excess is valid and protein-level confirmation would be beneficial, conducting these additional analyses would exceed the current scope of our Communication. Given the recommendation to publish this work as a Communication, we have focused on summarizing our findings at the RNA level.
Comments #3: In line 215, T4 (1.5 μg of T4/100 g body weight), what is the basis for the concentration of T4 injection? In the body, T4 is mainly used as a reserve form of thyroid hormone, while T3 has a more direct and active effect on cells. So did the author consider the relationship between T3 and T4?
Response #3: The dosing concentrations were determined based on previous reports. We have added the original study as reference 42.
Comments #4: Hormones have negative feedback mechanisms, and hormone regulation can be regulated by multiple hormones. Did the author study or exclude the interference of other hormones on SGLT1?
Response #4: It is known that intestinal SGLT1 is regulated by dietary carbohydrate content and exhibits a diurnal rhythm. In this experiment, there were no differences in diet between the SHAM and T4 administration groups or between the +/+ and mut/mut groups. Furthermore, experiments were conducted at consistent times to minimize the influence of hormones affected by circadian rhythms.
Round 2
Reviewer 1 Report
Comments and Suggestions for Authors
The reviewer appreciates the authors' efforts to try addressing her previous comments. However she finds the revised version of the manuscript not sufficiently improved in term of data presentation and organization.
With regard to previous reviewer’s COMMENT #1: The qRT-PCR relative expression analysis of SGLT1 in the small intestine (Figure 1) was normalized against a single housekeeping gene, which is somewhat unusual. Have the authors tested other housekeeping genes to confirm stability of GAPDH under the treatment conditions? the authors probably misunderstood the requirement. It is necessary to use almost two different housekeeping genes to confirm the results of qRT-PCR. Thus, in addition to the GAPDH normalization, the authors should normalize the expression of target genes against another housekeeping gene, for example alpha Tubulin, beta Actin, ribosome small subunit 18S, Cyclophin-A or similar.
With regard to previous reviewer’s COMMENT #2: How many animals (mice and/or rats) were used in the experimental procedures? No information is provided in Material and Methods section. For a better interpretation of the data, column graphs should be replaced by dot-plot graphs to identify the number of individual animals involved in the experimental groupsthe authors replaced the previous column graphs with dot-plot graphs. However, the dot-plot graphs chosen are not correct since parameters as mean/median/standard deviation are not indicated and visible. Thus, the authors should change this representation with column scatter dot-plot graphs. Furthermore, it is necessary add information about the number of animals involved in the experimental procedures, since the authors did not upgrade the Material and Methods section in spite the reviewer’s request.
With regard to previous reviewer’s COMMENT #3: Although the authors reported that treatment with T4 increases SGLT1 expression levels in the small intestine of rats, the data shown in Figure 2F do not appear to confirm a significant SGLT1 upregulation following hyperthyroid treatment. This data, not clearly evident from the qRT-PCR analyses, should necessarily be confirmed by protein expression. The same goes for the TRbD337T knock in mouse model the authors assessed a significant increase in SGLT1 mRNA levels (although the graph in figure 2F shows only a slight increase from 1.0 to 1.4). Considering the points in the dot-plot graph, it seems that there are no differences between sham and T4 groups. For this reason, the reviewer retains that qRT-PCR analyses, should necessarily be confirmed by protein expression. The fact that the authors were recommend publishing this article as a Communication does not represent a valid reason to underestimate concepts that are important and necessary to be clarified before a publication. The same for previous reviewer’s COMMENT #5:In vivo data should be corroborated by in vitro analyses using a small intestine cell line, treated with T4 (to mimics hyperthyroid conditions) and tT3 (to mimics hypothyroid conditions) to assess transcriptomic and protein expression of SGLT1. Furthermore, rescue experiments using a shRNA or siRNA against SGLT1 are necessary to explore the mechanistic regulatory pathway governing the thyroid hormones effects under postprandial hyperglycemia condition.
In conclusion, the reviewer does not approve the manuscript for the publication in the current state since a major revision is required for the reaching of the expected quality standards necessary for a publication. The editor has the final say on whether or not to accept the manuscript in its present condition.
Reviewer 2 Report
Comments and Suggestions for Authors
In this article entitled "Thyroid hormones regulate postprandial glucose metabolism by regulating SGLT1 expression in the small intestine", the authors used T4-administered rats and TRβΔ337T knock mouse models to study the effects of thyroid hormones on SGLT1 expression and its effects on postprandial glucose metabolism. In addition, the existing problems in this manuscript have been revised and supplemented, which has certain research significance and innovation, and we agree to accept it in its current form.